# Bub3 reads phosphorylated MELT repeats to promote spindle assembly checkpoint signaling

Ivana Primorac[1], John R Weir[1†], Elena Chiroli[2†], Fridolin Gross[2], Ingrid Hoffmann[1], Suzan van Gerwen[1], Andrea Ciliberto[2], Andrea Musacchio[1,3]*

[1]Department of Mechanistic Cell Biology, Max Planck Institute of Molecular Physiology, Dortmund, Germany; [2]IFOM—The FIRC Institute of Molecular Oncology, Milan, Italy; [3]Centre for Medical Biotechnology, Faculty of Biology, University Duisburg-Essen, Essen, Germany

**Abstract** Regulation of macromolecular interactions by phosphorylation is crucial in signaling networks. In the spindle assembly checkpoint (SAC), which enables errorless chromosome segregation, phosphorylation promotes recruitment of SAC proteins to tensionless kinetochores. The SAC kinase Mps1 phosphorylates multiple Met-Glu-Leu-Thr (MELT) motifs on the kinetochore subunit Spc105/Knl1. The phosphorylated MELT motifs (MELT$^P$) then promote recruitment of downstream signaling components. How MELT$^P$ motifs are recognized is unclear. In this study, we report that Bub3, a 7-bladed β-propeller, is the MELT$^P$ reader. It contains an exceptionally well-conserved interface that docks the MELT$^P$ sequence on the side of the β-propeller in a previously unknown binding mode. Mutations targeting the Bub3 interface prevent kinetochore recruitment of the SAC kinase Bub1. Crucially, they also cause a checkpoint defect, showing that recognition of phosphorylated targets by Bub3 is required for checkpoint signaling. Our data provide the first detailed mechanistic insight into how phosphorylation promotes recruitment of checkpoint proteins to kinetochores.

*For correspondence: andrea.musacchio@mpi-dortmund.mpg.de

†These authors contributed equally to this work

Competing interests: The authors declare that no competing interests exist.

## Introduction

Protein kinases are ubiquitous and almost invariably crucial components of cellular signaling networks (*Huse and Kuriyan, 2002*; *Ubersax and Ferrell, 2007*). Kinases transfer a high-energy phosphate group from ATP to the side chains of serine, threonine, tyrosine, and more rarely those of other residues (*Hunter, 2012*). The addition of phosphate groups can modify the activity of a target protein directly or indirectly through the modification of its pattern of physical interactions, which in turn might modify the target's activity, localization, or stability. Phosphorylation is usually a transient state that is reversed by the action of phosphatases. The transient nature of phosphorylation makes it ideally suited for use in signaling networks, where rapid activation and inactivation of defined substrates is important for the networks' ability to toggle between alternative states.

Phosphorylation plays a crucial role also in the spindle assembly checkpoint (SAC), a signaling network required for accurate chromosome segregation during cell division (*Lara-Gonzalez et al., 2012*; *Foley and Kapoor, 2013*). Checkpoint control creates a dependency between the mechanical aspects of cell division—the complex physical interaction of chromosomes with the mitotic spindle–and the timing of cell cycle progression. To prevent premature sister chromatid separation and mitotic exit in cells whose chromosomes have not yet attained bipolar attachment on the mitotic spindle, the SAC targets the cell cycle machinery required for the metaphase-to-anaphase transition (*Lara-Gonzalez et al., 2012*; *Foley and Kapoor, 2013*).

**eLife digest** The cell cycle is the process by which a cell divides to produce two near-identical daughter cells. Two crucial parts of the cell cycle are the duplication of the chromosomes in the original cell, and the segregation of these chromosomes between the two daughter cells. These and other parts of the cell cycle are strictly regulated to prevent errors, which can lead to cancer and other diseases.

After chromosome duplication has taken place, the pairs of identical chromosomes, known as sister chromatids, remain tightly bound to each other. These sister chromatids line up in the middle of the cell, with protein filaments called microtubules connecting them to a bipolar structure called the spindle. For the cell to divide correctly, the sister chromatids in each pair must be connected to opposite poles of the spindle. A signalling network known as the spindle assembly checkpoint (SAC) ensures that the sister chromatids have enough time to line up correctly and to correct possible problems. Once everything is in place, the SAC releases its 'break', and the microtubules then pull the sister chromatids away from each other. This way, each daughter cell receives the same complement of chromosomes that was present in the mother cell.

The microtubules are not directly attached to the sister chromatids but to protein complexes called kinetochores that assemble on each sister chromatid. In particular, each microtubule binds to a very large protein complex called the KMN network. Knl1, which is part of this network, recruits two SAC proteins–Bub1 and Bub3–to the kinetochore. It is known that a phosphate group is added to Knl1 when the SAC is active, and that Knl1 can only recruit Bub1 and Bub3 after it has been phosphorylated. However, the details of the interactions between Knl1, Bub1 and Bub3 are not understood, and it is not clear whether these interactions are essential for the SAC.

Now Primorac et al. have shown that Bub3 binds directly to Knl1 through a region that contains multiple MELT motifs (where M, E, L and T are all amino acids), and that this interaction only happens if these 'MELT repeats' have been phosphorylated. Moreover, once bound to the Knl1, Bub3 then recruits Bub1 to the kinetochore. By showing that the recognition of phosphorylated Knl1 by the Bub1-Bub3 complex has a central role in the spindle assembly checkpoint, these results highlight the importance of phosphorylation as a way of regulating the timing of events during the cell cycle.

Work in *Saccharomyces cerevisiae* originally identified several checkpoint components, including Bub1, Bub3, Mad1, Mad2, Mad3/BubR1, and Mps1 (*Hoyt et al., 1991*; *Li and Murray, 1991*; *Hardwick et al., 1996*), which were later found to be *de facto* ubiquitous in eukaryotes. Within this group, Bub1 and Mps1 are protein kinases. Together with all additional known checkpoint components, Bub1 and Mps1 become highly enriched at kinetochores between mitotic prophase and early prometaphase. Kinetochores are large protein assemblies, built on chromosomal loci known as centromeres. They bind directly to spindle microtubules to ensure the equational and reductional division of chromosomes during mitosis and meiosis, respectively (*Santaguida and Musacchio, 2009*). The dynamic interplay between kinetochore attachment to microtubules and checkpoint control is crucial for life in metazoans, but it remains disappointingly poorly understood.

Among the targets of the Mps1 kinase activity is a kinetochore protein named Spc105/Knl1 (also known as Spc7, Blinkin, CASC5 in different organisms) (*London et al., 2012*; *Shepperd et al., 2012*; *Yamagishi et al., 2012*). Spc105/Knl1 is the largest subunit of a 10-subunit assembly, the KMN network, which is believed to provide the main site of attachment of kinetochores to microtubules (*Figure 1A,B*) (reviewed in *Santaguida and Musacchio, 2009*). Within Spc105/Knl1, Mps1 phosphorylates at least a subset of an array of motifs that are generally referred to as 'MELT' and that conform to the consensus M-[E/D]-[L/I/V/M]-T (*Figure 1C*; we indicate as MELT$^P$ the phosphorylated form of a MELT motif). The presence of multiple MELT repeats is an essentially invariant feature of Spc105/Knl1 in evolution (*Cheeseman et al., 2004*).

How the phosphorylation on MELT motifs is interpreted by downstream components of the checkpoint pathway is unclear. Bub1 and Bub3, a 7-bladed WD40-repeat β-propeller that is constitutively bound to Bub1 (*Figure 1A*), are robustly recruited to Spc105/Knl1 when the MELT repeats are phosphorylated (*London et al., 2012*; *Shepperd et al., 2012*; *Yamagishi et al., 2012*), in line with previous observations linking Mps1 kinase activity to kinetochore recruitment of Bub1 and Bub3 (*Vanoosthuyse et al., 2004*;

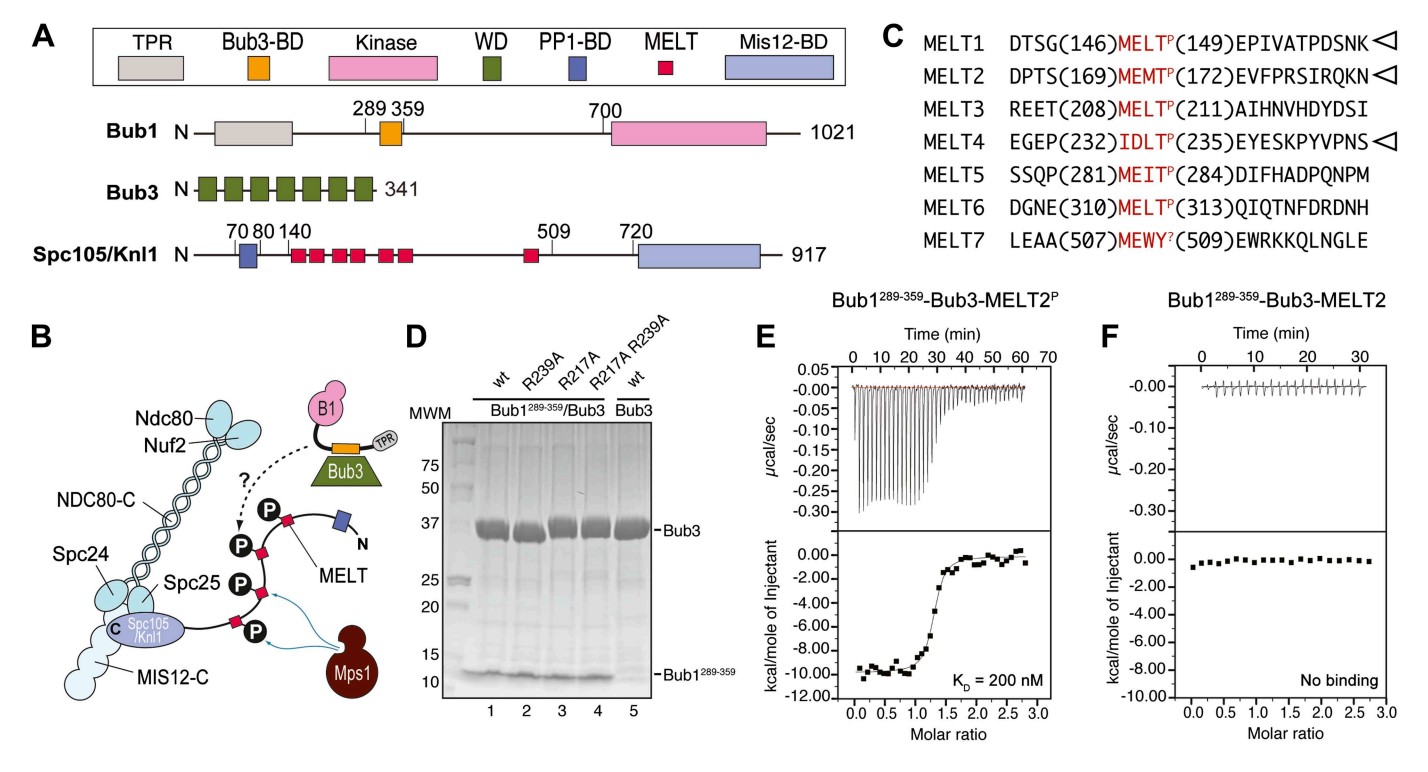

**Figure 1**. Reconstitution of the interaction of Bub1-Bub3 with MELT$^P$ motifs. (**A**) Schematic description of the domain and motif organization of the main players discussed in this paper. (**B**) The KMN network (shown in different tones of blue) consists of the Ndc80 complex (NDC80-C), the Mis12 complex (MIS12-C, also known as MIND complex), and Spc105/Knl1 (which also associates with Ydr532cp/Zwint, not shown here). Mps1 phosphorylates the MELT repeats of Spc105/Knl1 to promote the recruitment of the Bub1–Bub3 complex. A Bub3-binding domain of Bub1 is shown in orange. (**C**) Sequence of MELT repeats in Spc105/Knl1 of *S. cerevisiae*. Arrowheads indicate MELT repeats previously shown to be phosphorylated by Mps1 in vitro (**London et al., 2012**). The MELT motifs are shown in red. (**D**) Purified Bub3, Bub1$^{289–359}$–Bub3, and mutants thereof discussed in the text were separated by SDS-PAGE after purification. (**E**) Isothermal titration calorimetry (ITC) analysis of the interaction of Bub1$^{289–359}$–Bub3 with a synthetic peptide corresponding to the phosphorylated version of the second MELT$^P$ peptide (MELT2$^P$) shown in **C**. (**F**) ITC analysis, with the unphosphorylation version of the same peptide (MELT2), shows no binding.

The following figure supplements are available for figure 1:

**Figure supplement 1**. Additional calorimetry experiments.

*Vigneron et al., 2004*; *Kiyomitsu et al., 2007, 2011*; *Pagliuca et al., 2009*; *Schittenhelm et al., 2009*; *Maciejowski et al., 2010*; *Santaguida et al., 2010*; *Ito et al., 2011*; *Storchová et al., 2011*; *Heinrich et al., 2012*). However, whether Bub1 and Bub3 are sufficient for a tight interaction with MELT$^P$ repeats is currently unknown, and so is, therefore, the identity of the binding site for MELT$^P$ (*Figure 1B*).

Here, we show that Bub3 binds directly and with high affinity to MELT$^P$ motifs. The crucial determinants of this interaction are extremely well conserved in evolution and are required for a functional checkpoint. We discuss the recruitment mechanism of Bub1–Bub3 and its implications for checkpoint signaling. The constellation of Bub3 residues implicated in MELT$^P$ binding is perfectly conserved in the nucleoporin Rae1, suggesting that Rae1 might also be implicated in phosphopeptide binding.

## Results

### Reconstitution and quantitative analysis of the interaction of Bub1–Bub3 with P-MELT

Bub1 binds Bub3 through a conserved Bub3-binding domain (*Taylor and McKeon, 1997*) that is often also referred to as GLEBS motif (*Bailer et al., 1998*; *Wang et al., 2001*) (*Figure 1A*). The Bub3-binding

domain of Bub1 is necessary for kinetochore recruitment of Bub1. When expressed in isolation in human cells, this region of Bub1 is sufficient to mediate robust kinetochore recruitment of Bub1, albeit at partly reduced levels compared to constructs that also include the N-terminal TPR domain of Bub1 (*Taylor and McKeon, 1997*; *Vanoosthuyse et al., 2004*; *Klebig et al., 2009*; *Krenn et al., 2012*). The minimal region of human Bub1, capable of mediating kinetochore targeting, consists of residues 209–270 (equivalent to residues 289–359 of Bub1 in *S. cerevisiae*). Additional deletions of this segment prevent kinetochore binding (*Krenn et al., 2012*). Because the minimal kinetochore recruitment domain of Bub1 coincides with the Bub3-binding domain, and because it is known that Bub1 and Bub3 reinforce each other in kinetochore localization (*Taylor and McKeon, 1997*; *Taylor et al., 1998*; *Sharp-Baker and Chen, 2001*; *Millband and Hardwick, 2002*; *Gillett et al., 2004*; *Kadura et al., 2004*; *Vanoosthuyse et al., 2004*; *Rischitor et al., 2006*; *Logarinho et al., 2008*; *Klebig et al., 2009*; *Windecker et al., 2009*; *Krenn et al., 2012*; *London et al., 2012*; *Shepperd et al., 2012*; *Yamagishi et al., 2012*), it is expected that Bub1 and Bub3 cooperate in the mechanism of kinetochore recruitment.

In vitro reconstitution with recombinant purified material is often crucial to address the molecular mechanism of protein interactions. Thus, we attempted the reconstitution of the phosphorylation-dependent recruitment of Bub1–Bub3 to kinetochores. We chose to work with proteins from *Saccharomyces cerevisiae* because the Bub1–Bub3 complex has already been reconstituted in this organism (*Larsen et al., 2007*) and also because the identity of phosphorylated MELT repeats in ScSpc105 (we will refer to ScSpc105 as Spc105/Knl1 for the remains of this work) is known from previous work (*London et al., 2012*) (*Figure 1C*).

We generated recombinant ScBub1[289–359]–Bub3 (where the Bub1 segment is equivalent to the minimal kinetochore targeting region of human Bub1 [*Krenn et al., 2012*]) by bacterial co-expression and purified it to homogeneity (*Figure 1D*). To assess whether Bub1–Bub3 binds directly to MELT[P] sequences, we tested its ability to bind MELT sequences in quantitative isothermal titration calorimetry (ITC) binding experiments. 19-residue synthetic peptides encompassing the sequences of the second and fourth MELT[P] motifs of Spc105/Knl1 (indicated as MELT2 and MELT4, respectively), each flanked by four and eleven residues on the N- and C-terminal ends, respectively, were tested (*Figure 1C*). ScBub1[289–359]-Bub3 bound the MELT2[P] and MELT4[P] peptides with dissociation constants (KD) of 200 nM (*Figure 1E*) and 1.3 µM, respectively (*Figure 1—figure supplement 1*). Remarkably, no binding was observed with the non-phosphorylated versions of the MELT2 and MELT4 peptides (*Figure 1F* and *Figure 1—figure supplement 1*), indicating exquisite selectivity for the phosphorylated MELT motifs. Conversely, Bub3 did not show any binding affinity for an unrelated phospho-peptide (*Figure 1—figure supplement 1*). Thus, ScBub1[289–359]–Bub3 is sufficient for the reconstitution of tight interactions with two MELT[P] peptides in vitro that recapitulate a salient feature of this interaction, its dependency on phosphorylation.

## Structural analysis of the ScBub1[289–359]-Bub3-MELT[P] ternary complex

We crystallized the ScBub1[289–359]–Bub3-MELT2[P] ternary complex and determined its structure by X-ray crystallography to a resolution of 1.9 Å by molecular replacement with ScBub1[315–356]–Bub3 as a search model (PDB ID 2I3S; *Larsen et al., 2007*) (*Figure 2A–B*). The model extends to the two ternary complexes in the asymmetric unit and was refined to a 'free' R-factor ($R_{free}$) of 19.2%, with excellent stereochemical parameters (*Table 1*). The two trimers in the asymmetric unit are very similar, and their salient features can be described essentially equivalently.

Each blade of the 7-bladed Bub3 β-propeller consists of four β-strands, with the innermost and outermost strands referred to as βA and βD, respectively. Most intra- and inter-blade loops in Bub3 are short, giving rise to a rather regular toroid. The two notable exceptions are the βD5-βA6 and βB7-βC7 loops (*Figure 2A–C*), both of which interact extensively with Bub1. As shown previously (*Larsen et al., 2007*), Bub1 meanders on the top surface of the Bub3 β-propeller (defined as the surface that contains the βD–βA loops that connect consecutive blades). The fragment of ScBub1 contained in our crystals, however, is 29 residues longer (residues 289–315) at its N-terminus relative to the one previously co-crystallized with ScBub3 (*Larsen et al., 2007*) (*Figure 2—figure supplement 1*). Residues in this extension contribute to the formation of a β-hairpin (β1-β2, *Figure 2A and 2D*), which pairs, via β2, with a β-hairpin within the extended βD5-βA6 loop of Bub3. Together, the β-hairpins from Bub1 and Bub3 form a joint 4-stranded β-sheet that creates a 'roof' on the MELT[P] peptide.

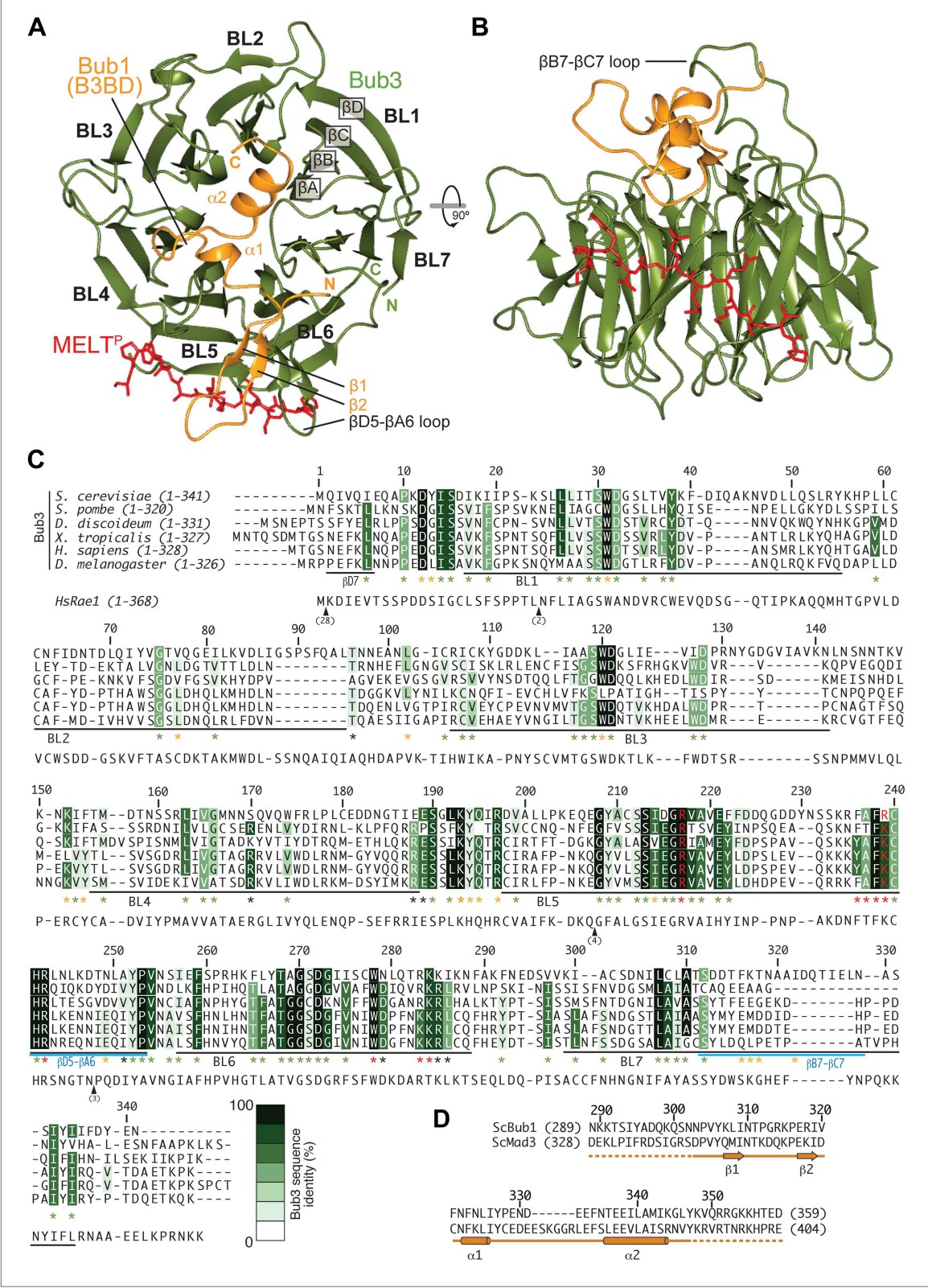

**Figure 2**. Structure and conservation of the Bub1–Bub3 complex. (**A**) Top view of the Bub1[289–359]–Bub3-MELT[P] ternary complex. N and C indicate the N- and C-terminus, respectively. (**B**) Side view of the ternary complex. (**C**) Sequence alignment of Bub3 from the indicated species. The presented alignment was extracted from a much larger alignment consisting of more than 40 Bub3 sequences from distant eukaryotes (**Vleugel et al., 2012**). The indicated

*Figure 2. Continued on next page*

*Figure 2. Continued*
levels of conservation were derived from the larger alignment. Green asterisks indicate residues predicted, on structural ground, to be important for the stability of the Bub3 propeller. Orange asterisks point to residues that interact with Bub1. Red asterisks point to residues that interact with the MELT[P] peptide. Black asterisks point to conserved residues of uncertain function. BL = blade. (**D**) Mapping of secondary structure elements on the sequence of the Bub3-binding domain of Bub1.
The following figure supplements are available for figure 2:

**Figure supplement 1**. Comparison of Bub3–Bub1 and Bub3–Mad3 structures with Rae1-Nup98.
**Figure supplement 2**. Composite omit maps of the region corresponding to the phospho-MELT peptide.

## The Bub3 propeller binds MELT[P] and defines a novel interaction mode

Bub3 plays a dominant role at the interface with the MELT2[P] peptide (*Figures 2 and 3*). The latter (for which there is excellent electron density between residues 166–176 [*Figure 2—figure supplement 2*]) docks on blades 4–6 of the Bub3 β-propeller, with its main chain oriented almost orthogonally to the vertical axis of the propeller's toroid. This docking mode, which is unprecedented in β-propeller-peptide interactions (*Figure 3—figure supplements 1 and 2*), is accompanied by the formation of at least five hydrogen bonds between the main chain atoms of the peptide and of the βD-strands of blades five and six and of the βD4-βA5 loop (*Figure 3A*).

The side chains of the MEMT[P] motif are also extensively involved in the interaction with Bub3. The binding site on Bub3 is essentially bipartite, with a highly hydrophobic 'south' interface interacting with the hydrophobic side chains of Met169[MELT] and Met171[MELT], and a highly positively charged 'north' interface contacting the acidic side chains of Glu170[MELT] and P-Thr172[MELT] (*Figure 3B*) Specifically, at the south end, the side chains of Met169[MELT] and of Met171[MELT] are embedded in a deep hydrophobic pocket lined up by the side chains of Phe236[Bub3], Phe238[Bub3], Trp278[Bub3], and by the aliphatic portion of the side chain of Arg283[Bub3] (*Figure 3C*). At the north end, the side chains of Glu170[MELT] and P-Thr172[MELT] face Arg217[Bub3], Arg239[Bub3], and Arg242[Bub3], which are therefore ideally positioned to compensate the negative charge of the phosphopeptide (*Figure 3D*, *Figure 3—figure supplement 3*) and are at the core of a complex network of hydrogen bonds that also engages Glu317[Bub1]. Within this array of residues, Arg242[Bub3], in the βD5-βA6 loop, faces Glu170[MELT] and has additional stabilizing effects on the side chains of Arg217[Bub3] and Arg239[Bub3], which face P-Thr172[MELT] (*Figure 3D*). Taken together, these interactions explain the positive discrimination by Bub1[289–359]–Bub3 for phosphorylated versions of the MELT peptides. Finally, at the 'west' end of the binding site, the aromatic side chain of Phe175[MELT2] (+3 position relative to P-Thr) stacks against the side chain of Lys193[Bub3], which is held in position by the side chain of Tyr194[Bub3] (not shown). Both Bub3 residues are exposed and invariable in evolution (*Figure 2C*), suggesting that they play an important functional role, but there is no strong preference for phenylalanine or other hydrophobic residues in the sequence of MELT repeats at the +3 position (*Figure 1C*). It is also possible that Lys193[Bub3] and Tyr194[Bub3] are required to stabilize the interaction with Bub1, whose Ile309[Bub1], Ile319[Bub1], and Phe323[Bub1] are in direct van der Waals contact with the side chain of Tyr194[Bub3] (not shown).

## Mutations on Bub1–Bub3 affect interaction with MELT[P] in vitro

A plot of sequence conservation on the surface of the Bub1–Bub3 complex (*Figure 3E–H*) shows an extreme concentration of conserved residues at the interface with the Spc105/Knl1 peptide. The level of conservation at this site even exceeds the conservation of Bub1-binding residues (*Figure 2C*). Thus, binding to phosphorylated sequences is a crucial property of Bub3. Overall, the pattern of sequence conservation strongly suggests that binding to MELT[P] and Bub1 might be the only two widely conserved functions of Bub3.

Because the presence of a phosphate on Thr172[Spc105/Knl1] is essential for high-affinity binding of ScBub1[289–359]–Bub3 to MELT[P] peptides (*Figure 1*), we concentrated our mutational analysis on two residues that are directly implicated in the recognition of the peptide's phospho-threonine, Arg217[Bub3] and Arg239[Bub3]. Positively charged residues are invariant at these positions of the Bub3 alignment (*Figure 2C*). We generated single or double alanine mutants of Arg217[Bub3] and Arg239[Bub3] in the context of

**Table 1.** Data collection and refinement statistics

| Data collection | |
| --- | --- |
| Wavelength (Å) | 1.21 |
| Resolution range (Å) | 46.8–1.95 (2–1.95) |
| Space group | C2 |
| Unit cell | a = 138.7; b = 57.9; c = 118.7; α = 90 β = 102.5 γ = 90 α = γ = 90° β = 102.5° |
| Total reflections | 438568 |
| Unique reflections | 66488 |
| Multiplicity | 6.6 (6.3) |
| Completeness (%) | 98.70 (88.81) |
| Mean I/sigma (I) | 14.15 (3.09) |
| Wilson B-factor | 26.16 |
| $R_{sym}$ | 0.069 (1.023) |
| CC(1/2)* | 99.9 (87.3) |
| **Refinement** | |
| R-factor | 0.1726 (0.2552) |
| R-free | 0.1916 (0.2864) |
| Number of atoms | 6412 |
| Macromolecules | 6052 |
| Metal ions | 2 |
| Water | 358 |
| Protein residues | 775 |
| Average B-factor | 36.40 |
| Macromolecules | 36.00 |
| Solvent | 42.20 |
| **Geometry** | |
| RMS (angles, Å) | 0.89 |
| RMS (bonds, °) | 0.006 |
| Ramachandran favored (%) | 97 |
| Ramachandran outliers (%) | 0 |
| MolProbity score† | 1.29 (99th percentile) |

Statistics for the highest-resolution shell are shown in parentheses.
*Percentage of correlation between intensities from random half-datasets (**Karplus and Diederichs, 2012**).
†MolProbity score combines the clashscore, rotamer, and Ramachandran evaluations into a single score, normalized to be on the same scale as X-ray resolution (**Chen et al., 2010**).

ScBub1[289–359]–Bub3 and tested their binding affinity for the MELT2[P] peptide by ITC. Importantly, the mutant complexes were expressed and purified essentially like the wild type complex and did not suffer obvious losses of stability (**Figure 1D**).

Replacement of Arg217 or Arg239 with alanine caused a 13- to 25-fold reduction in the binding affinity for the MELT2[P] peptide, with $K_D$s of 2.7 μM and 5 μM for the R217A and R239A mutants (**Figure 4A,B**), respectively, compared with 200 nM for the wild type interaction (**Figure 1E**). Thus, neither arginine side chain is completely indispensable for binding, but each is required for high-affinity binding. When the mutations were combined in the Bub3[R217A–R239A] double mutant, no significant residual binding to the MELT[P] peptide was observed (**Figure 4C**). Collectively, the mutational analysis is in line with the observation that the binding of ScBub1[289–359]–Bub3 to the MELT2[P] peptide is exquisitely phosphorylation-sensitive.

## Role of Bub1

The majority of residues involved in the interaction with the MELT[P] motifs of Spc105/Knl1 are located in Bub3, indicating that the latter plays the prominent role in kinetochore recruitment of Bub1. However, as already anticipated in the Introduction, several lines of evidence indicate that Bub1 contributes to this interaction (Discussion). To test this possibility formally, we measured the binding of purified Bub3 (**Figure 1D**) to the MELT2[P] peptide in the absence of Bub1. Remarkably, we observed a 10-fold reduction in the binding affinity of Bub3 for the MELT2[P] peptide ($K_D$ = 2 μM; **Figure 4D**) compared to the Bub1–Bub3 complex (**Figure 1E**), indicating that Bub1 does indeed positively contribute to the interaction.

Such function of Bub1 is probably exerted primarily through its structuring effects on the 4-stranded β-sheet 'roof' that dominates the peptide-binding region of Bub3 and which restrains the position of the positively charged residues in the 'north' area of the MELT[P]-binding site (**Figure 4E**). Additionally, we observe that Arg314[Bub1], in the β1-β2 loop, contributes to the interaction with the phosphate group of

P-Thr172[Spc105/Knl1] (the interaction, however, is only observed in one of the two complexes in the asymmetric unit, and in proximity of a crystal contact) and is therefore directly engaged in the interaction with the MELT2[P] peptide. In ITC measurements, we observed a ~fourfold reduction in the binding affinity of the Bub3-Bub1[R314A] mutant for the MELT2[P] peptide, in agreement with a role of Arg314[Bub1] is in the interaction of the Bub1-Bub3 complex with MELT2[P] (**Figure 4F**). Similarly, Bub3–Bub1[R314A] bound the MELT4[P] with ~threefold decreased affinity compared to wild type Bub3–Bub1 (data not shown).

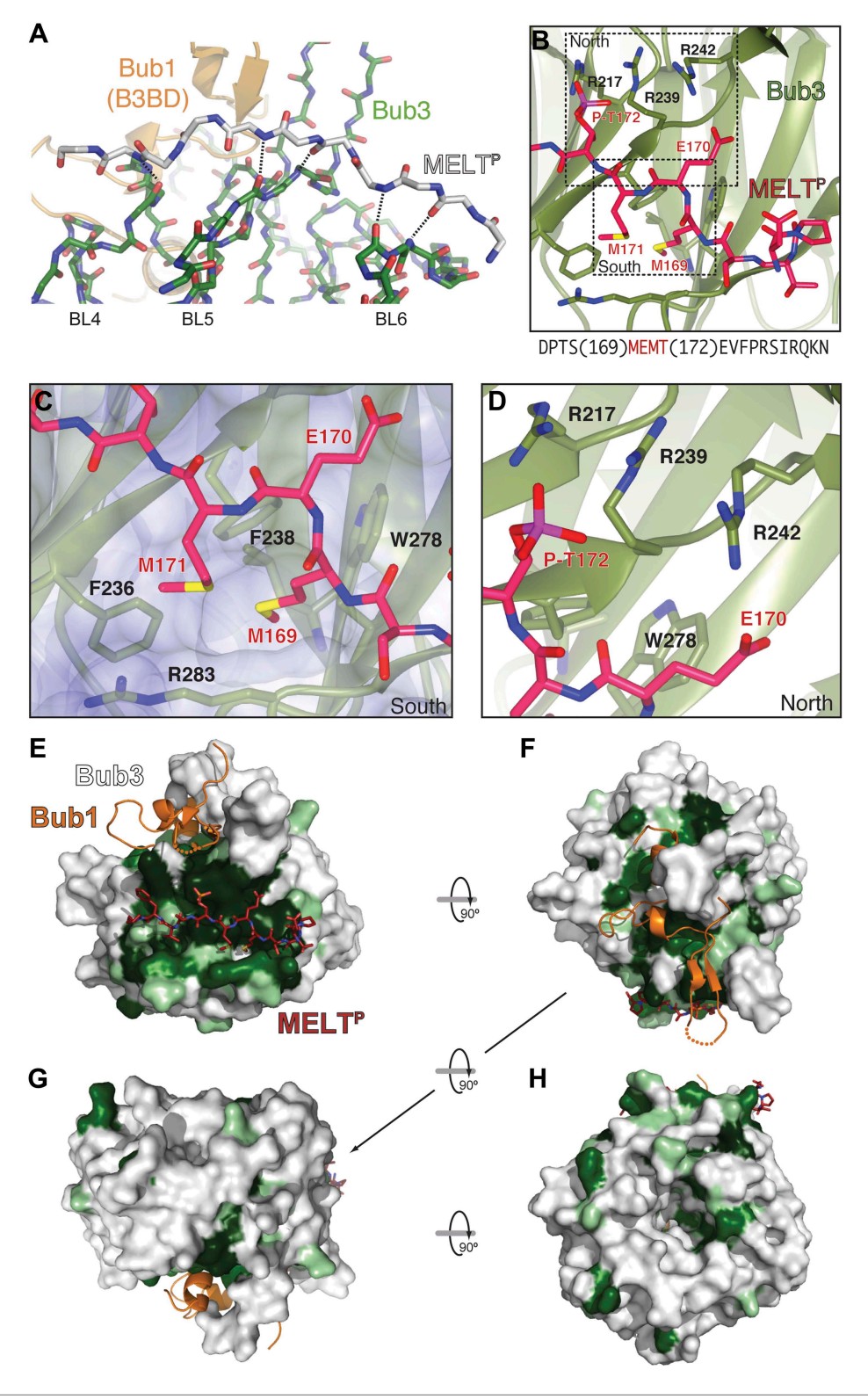

**Figure 3**. The interface between Bub1–Bub3 and MELT[P]. (**A**) The MELT[P] peptide (here shown with carbon atoms in light gray color) orients transversally to the blades but its main chain amide and carbonyl groups form several hydrogen bonds with the main chain of the outermost strands of three consecutive blades, blades four to six. (**B**) Details of the interaction around MELT[P] sequence. The boxed regions are enlarged in panels **C** and **D**. (**E–H**) Surface representation of
*Figure 3. Continued on next page*

*Figure 3. Continued*

sequence conservation (resulting from the alignment discussed in the legend of *Figure 2C*) shows a dramatic concentration at the interface with MELT$^P$.

The following figure supplements are available for figure 3:

**Figure supplement 1**. A collection of modes of ligand binding by β-propellers.

**Figure supplement 2**. Mode of binding of phosphopeptides from Cyclin E and β-catenin to the β-propellers of Fbw7 and β-TrCP.

**Figure supplement 3**. Electrostatics on the Bub3 surface at the MELT$^P$ interface.

## Effect of Bub3 mutations on Bub3 and Bub1 localization to kinetochores

Next, we asked if mutations in Bub3 that prevent its interaction with MELT2$^P$ motifs in vitro also affected its recruitment to kinetochores. To this end, we inserted three copies in tandem of the coding sequence for mCherry in frame at the 3′ end of the coding sequence of *S. cerevisiae BUB3* or of the *bub3$^{R217A–R239A}$* mutant. The transgenes were inserted at the TRP1 locus of a *bub3Δ* strain. The resulting strains were viable and expressed similar levels of wild type or mutant Bub3 (*Figure 5—figure supplement 1*).

To assess if the Bub3-mCherry localized to kinetochores, we tested its co-localization with Mtw1, a subunit of the MIS12 kinetochore complex (MIS12-C, also known as MIND complex, *Figure 1B*). Unsynchronized *S. cerevisiae* cells expressing tagged versions of Bub3 and of the kinetochore subunit Mtw1 (Bub3-mCherry and Mtw1-GFP) were imaged in a flow cell by wide-field fluorescence microscopy. Bub3-mCherry appeared to co-localize with Mtw1 shortly before budding and until approximately metaphase (*Figure 5A*, *Video 1*), in agreement with a previous study (*Gillett et al., 2004*). This behavior of Bub3-mCherry was formalized, for each video frame, through computation of a 'localization index' whose peaks coincide with kinetochore recruitment (*Figure 5B*, 'Materials and methods', *Figure 5—figure supplement 2* for details). Thus, as shown previously (*Gillett et al., 2004*), Bub3-mCherry localizes to kinetochores during each cell cycle in unperturbed *S. cerevisiae* cells. In contrast to Bub3-mCherry, Bub3$^{R217A–R239A}$-mCherry never co-localized with Mtw1-GFP during the cell cycle, indicative of defective kinetochore recruitment (*Figure 5C–D*, *Video 2*). This behavior of Bub3$^{R217A–R239A}$-mCherry agrees with the inability of the recombinant Bub3 mutant to interact with phosphorylated MELT repeats in vitro (*Figure 4C*).

Collectively, these results indicate that the integrity of the MELT$^P$ binding site of Bub3 is essential for its kinetochore recruitment. Because Bub1 interacts with Bub3, we asked if its pattern of kinetochore localization was similar to that of Bub3-mCherry (*Figure 5E*, *Video 3*). Indeed, the kinetochore localization indexes for Bub3-mCherry and of a Bub1–GFP construct peaked at the same time (*Figure 5F*). Thus, also Bub1–GFP localizes to kinetochores during an unperturbed cell cycle in *S. cerevisiae*.

Next, we tested if Bub1–GFP localized to kinetochores in cells expressing Bub3$^{R217A–R239A}$-mCherry in *bub3Δ* cells. Kinetochore localization of Bub1–GFP was completely suppressed in these cells (*Video 4*, *Figure 5G*), and the kinetochore localization index was correspondingly flat (*Figure 5H*). In summary, these observations provide a clear demonstration of the fact that the interaction of Bub3 with MELT$^P$ motifs is crucial for the kinetochore recruitment of the Bub1–Bub3 complex.

## Bub3 mutations disrupt the spindle assembly checkpoint

Next, we asked if the mutations in Bub3 that affect Spc105/Knl1 binding in vitro or in vivo also affect the ability of *S. cerevisiae* cells to activate the spindle checkpoint. Cells arrested in G1 with α-factor were released in the cell cycle in the presence of nocodazole to activate the spindle checkpoint. To assess checkpoint proficiency, we monitored the ability of cells to arrest in mitosis and to prevent re-replication, as well as lack of rebudding. Wild type cells and *bub3Δ* expressing Bub3-mCherry cells completed DNA replication at ~60 min after release from the G1 block in nocodazole and arrested as budded cells with 2C DNA content (*Figure 6*, panels A, C and E), indicative of a functional SAC. Conversely, *bub3Δ* cells and *bub3Δ* expressing Bub3$^{R217A–R239A}$-mCherry cells were unable to arrest, re-replicated their DNA, and re-budded, indicative of a disrupted SAC (*Figure 6*, panels B, D and E).

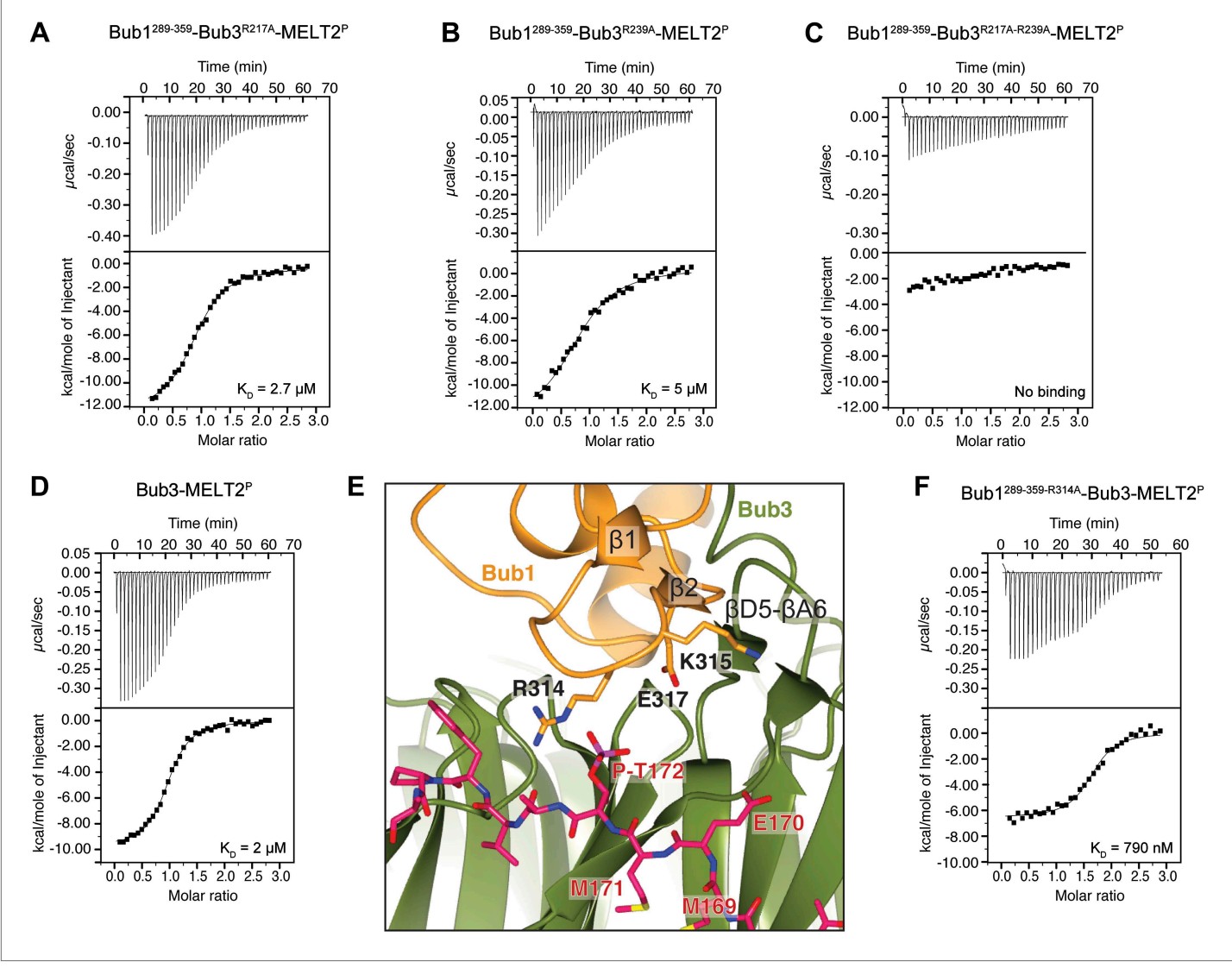

**Figure 4**. Biochemical validation of the interaction. (**A**) ITC analysis of the interaction of Bub1[289–359]–Bub3[R217A] with a synthetic peptide encompassing the MELT2[P] sequence. (**B–D**) ITC experiments with the MELT2[P] peptide and Bub1[289–359]–Bub3[R239A], Bub1[289–359]–Bub3[R217A–R239A] double mutant, and Bub3, respectively. (**E**) Close-up of the MELT[P] binding site indicating possible roles of the Bub3-binding motif of Bub1. The amino acid sequence of the Bub3-binding domain of Bub1 is reported in *Figure 2D*. (**F**) ITC experiment with the MELT2[P] peptide and the Bub1[289–359–R314A]–Bub3 complex.

These observations demonstrate that the spindle checkpoint is disrupted when the ability of Bub3 to interact with MELT[P] sequences is impaired.

## Discussion

Detailed information on the mechanisms of recruitment to, activation at, and release from kineto-chores of the checkpoint proteins has been missing. As a consequence, it has been hard to design targeted experiments aiming to dissect the role of specific binding interactions in the regulation of checkpoint proteins. Phosphorylation regulates many interactions at the kinetochore (*Lara-Gonzalez and Taylor, 2012*; *Foley and Kapoor, 2013*). In a few well-characterized cases, phosphorylation negatively regulates protein interactions at the kinetochore (e.g., *Cheeseman et al., 2006*; *DeLuca et al., 2006*; *Meadows et al., 2011*; *Rosenberg et al., 2011*). The kinase activity of Mps1, on the other hand, has a positive role in the recruitment of other checkpoint proteins to the kinetochore (*Lara-Gonzalez and Taylor, 2012*; *Foley and Kapoor, 2013*). The recent discovery that MELT[P] motifs are required for the recruitment of the Bub1–Bub3 complex represented an important advancement

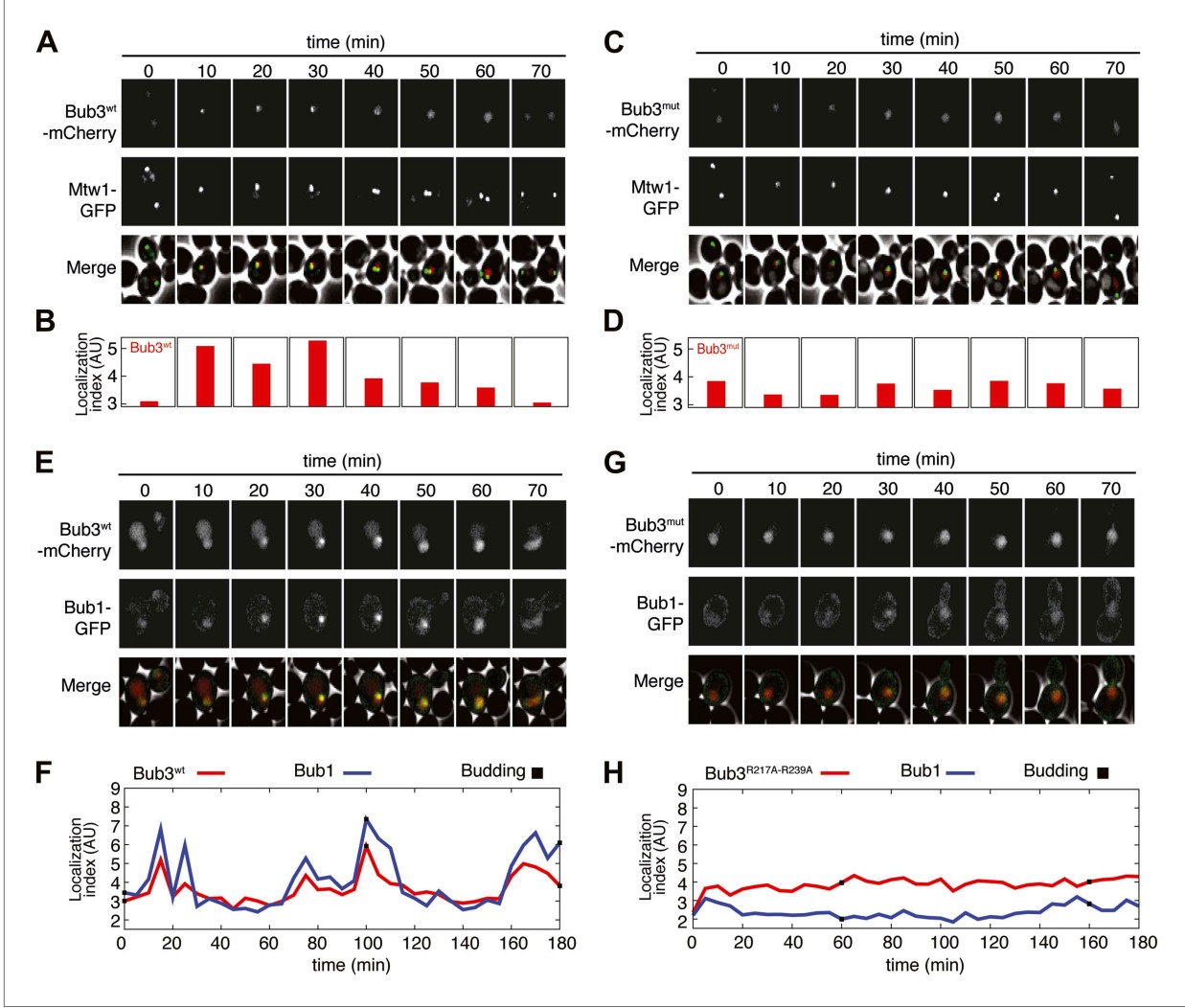

**Figure 5**. Mutant Bub3 does not localize to kinetochores and mislocalizes Bub1. (**A**) Live *S. cerevisiae* cells expressing wild type Bub3-mCherry and Mtw1-GFP where filmed to assess kinetochore localization of the fluorescent proteins (**Video 1**). Selected frames are shown. (**B**) A localization index was calculated as discussed in 'Materials and Methods'. High values of the index indicate recruitment of Bub3-mCherry to kinetochores. (**C**) As in panel **A**, but using cells expressing Bub3$^{R217A–R239A}$-mCherry. (**D**) Localization index for Bub3$^{R217A–R239A}$-mCherry. The localization index for the Bub3 mutant fluctuates around the value of 3.7, which we identify as corresponding to 'perfect delocalization' ('Materials and methods'). (**E**) Selected frames from **Video 3** demonstrating kinetochore localization of Bub3-mCherry and Bub1–GFP. (**F**) Peaks in the localization index indicate the timing of kinetochore recruitment of Bub3-mCherry and Bub1–GFP during subsequent cell cycles. Red and blue curves report localization of Bub3-mCherry and Bub1–GFP, respectively. The time of initiation of budding is marked by black squares. The diagram extends to three budding events. There is excellent correlation of the Bub1 and Bub3 signal, indicative of co-localization. (**G**) Bub3$^{R217A–R239A}$-mCherry does not localize to kinetochores (**Video 2**). Bub1–GFP fails to localize to kinetochores in Bub3$^{R217A–R239A}$-mCherry cells (selected frames from **Video 4**), in agreement with the role of Bub3 in kinetochore recruitment of Bub1 (**Gillett et al., 2004**). (**H**) The localization index for Bub3$^{R217A–R239A}$-mCherry and Bub1–GFP is flat, close the numerical value corresponding to delocalization in wild type cells.

The following figure supplements are available for figure 5:

**Figure supplement 1**. Expression levels of Bub3 and Bub3 mutants in bub3Δ *Saccharomyces cerevisiae's* cells.

**Figure supplement 2**. Validation of the localization index.

(**London et al., 2012**; **Shepperd et al., 2012**; **Yamagishi et al., 2012**). Here, we have taken a considerable step forward by identifying Bub3 as the MELT$^P$ reader and by probing its importance for checkpoint signaling.

Specifically, ours is the first detailed mechanistic description of how the phosphorylation of a kinetochore subunit promotes recruitment of downstream elements. Our studies identify Bub3 as a new

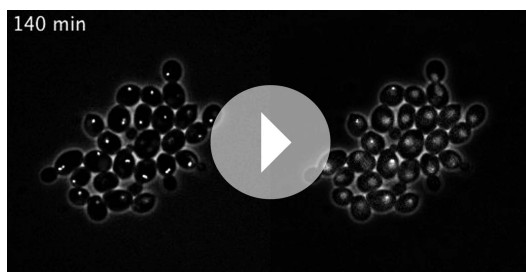

**Video 1**. Localization of Mtw1-GFP (left) and Bub3^wt^-mCherry (right) in replicating *S. cerevisiae's* cells.

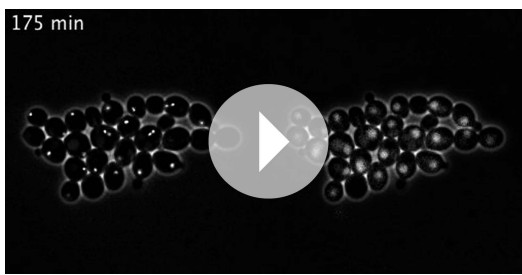

**Video 2**. Localization of Mtw1-GFP (left) and diffuse localization of Bub3^R217A–R239A^-mCherry (right) in replicating *S. cerevisiae's* cells.

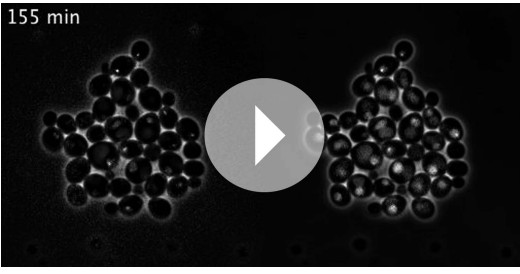

**Video 3**. Localization of Bub1-GFP (left) and Bub3^wt^-mCherry (right) in replicating *S. cerevisiae's* cells.

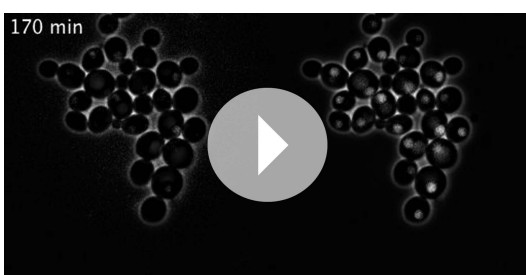

**Video 4**. Diffuse localization of Bub1–GFP (left) in replicating *S. cerevisiae's* cells expressing Bub3 ^R217A–R239A^_mCherry (right).

member of a family of protein domains involved in the recognition of phosphorylated sequence motifs, which includes, among others, the SH2, PTB, BRCT, FHA and polo-box domains (*Seet et al., 2006*; *Pawson and Kofler, 2009*). Interestingly, the structure of the complex of the GLEBS motif of the nucleoporin and proto-oncogene Nup98 with the WD40-repeat nuclear transport factor Rae1/Gle2 (PDB ID 3MMY) (*Ren et al., 2010*) is very closely related to that of the Bub3–Bub1 complex (*Figure 2C*, *Figure 2— figure supplement 1*). Most of the residues involved in MELT^P^ binding on Bub3 are perfectly conserved in Rae1 (*Figure 2C*), where they form a hydrophobic-basic bipartite interface that is almost indistinguishable from that in Bub3 (*Ren et al., 2010*). The striking conservation at this interface, and our realization that the interface is implicated in MELT^P^ binding, suggests that Rae1, or possibly its complex with Nup98, might also be a receptor for phosphorylated motifs.

The kinetochore levels of SAC proteins are dynamically regulated during the process of attachment of kinetochores to microtubules, with maximal levels being reached at unattached kinetochores and minimal levels being reached at metaphase (*Lara-Gonzalez et al., 2012*; *Foley and Kapoor, 2013*). It is plausible that such dynamic behavior reflects a requirement for specific steps of activation and inactivation of the checkpoint proteins at kinetochores in response to the status of kinetochore-microtubule attachment. For instance, forced retention of the Mad1-Mad2 'template' complex (*De Antoni et al., 2005*) at kinetochores results in a protracted checkpoint-dependent arrest despite all chromosomes being bipolarly aligned at the metaphase plate (*Gassmann et al., 2010*; *Maldonado and Kapoor, 2011*), indicating that removal of Mad1-Mad2— which is normally mediated by the Dynein-Spindly complex (*Griffis et al., 2007*; *Yamamoto et al., 2008*; *Chan et al., 2009*; *Barisic et al., 2010*; *Gassmann et al., 2010*)—is necessary for the inactivation of the checkpoint signal before anaphase.

The importance of kinetochore recruitment of Bub1–Bub3 in the spindle checkpoint, on the other hand, has been controversial (*Klebig et al., 2009*; *Vanoosthuyse et al., 2009*; *Windecker et al., 2009*; *Shepperd et al., 2012*; *Yamagishi et al., 2012*). Recently, however, it was shown that mutation of the phosphorylated Thr residue in the MELT repeats of Spc105/Knl1 prevents kinetochore recruitment of Bub1–Bub3 and results in a checkpoint defect in several species (*London et al., 2012*; *Shepperd et al., 2012*; *Yamagishi et al., 2012*). Formally, checkpoint deficiency in

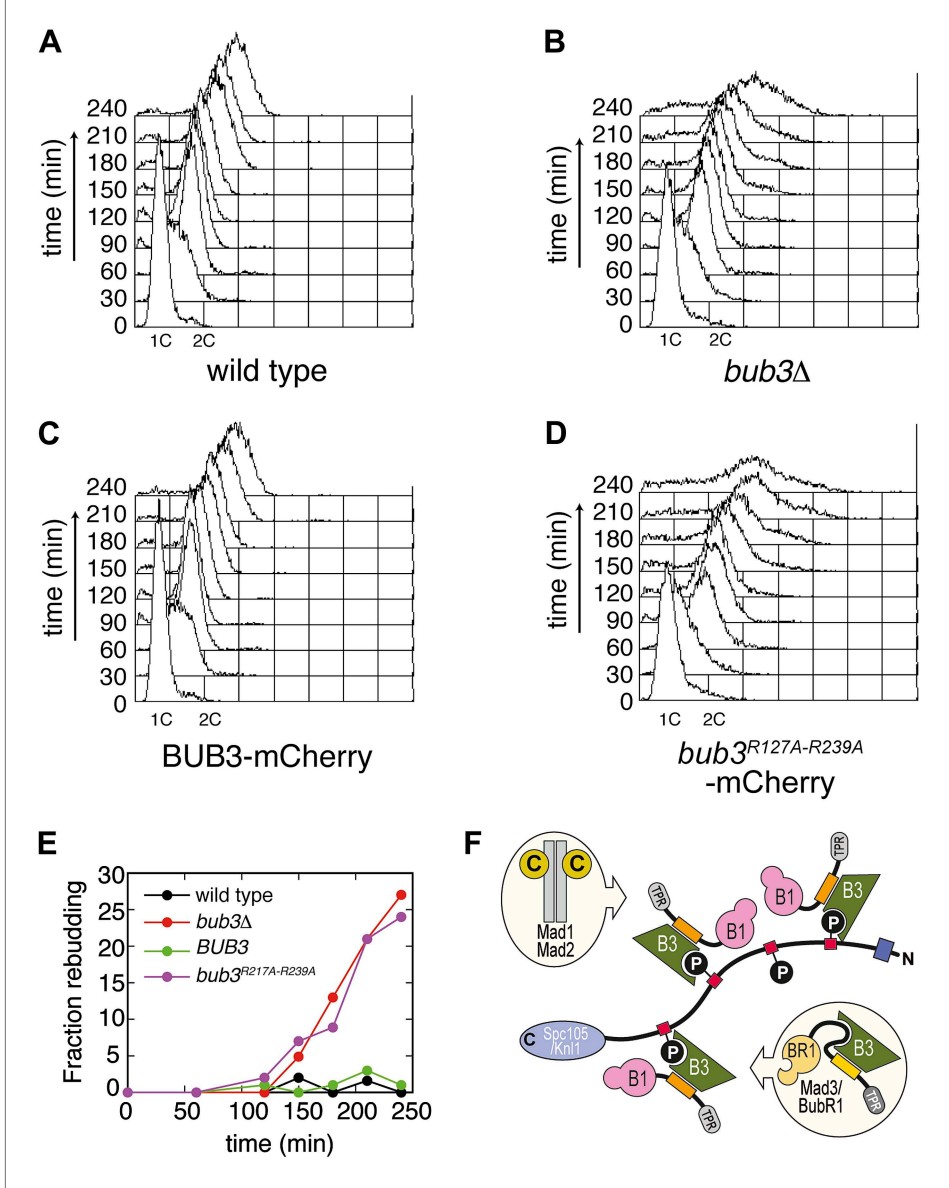

**Figure 6**. Mutant Bub3 cannot sustain the checkpoint. (**A**) G1-arrested wild type *S. cerevisiae* cells were released in the cell cycle in the presence of nocodazole. FACS analysis at the indicated time points shows that cells first undergo DNA replication and subsequently arrest with 2C DNA content, indicative of mitotic checkpoint arrest. (**B**) *bub3Δ* cells are checkpoint deficient, fail to arrest, and re-replicate they DNA. (**C**) A functional checkpoint is re-established upon expression of wild type Bub3 in *bub3Δ* cells. (**D**) Bub3[R217A–R239A] is unable to restore a functional checkpoint when expressed in *bub3Δ* cells. Panels **A**–**D** report experiments that were carried out at the same time and at least twice. (**E**) Re-budding in the presence of nocodazole was taken as an independent indication of checkpoint deficiency. Wild type cells, and *bub3Δ* cells reconstituted with wild type Bub3 were able to maintain the checkpoint arrest and did not re-bud during the time of observation. Conversely, *bub3Δ* cells and cells reconstituted with Bub3[R217A–R239A] re-budded, indicative of checkpoint failure. (**F**) The binding affinity of Bub1–Bub3 for individual MELT[P] is high. This predicts that multiple Bub1–Bub3 complexes may become bound to a single Spc105/Knl1 molecule. Mad3/BubR1 requires Bub1 for kinetochore recruitment, indicating that it is not able to target autonomously to kinetochores. Because Mad3/BubR1 is, like Bub1, constitutively bound to Bub3, it is plausible that Mad3/BubR1 suppresses the MELT[P]-binding activity of Bub3. Whether this occurs, and how, are purely speculative at this time.

the presence of mutants that prevent the phospohorylation of the MELT motifs might be due to the impaired binding of MELT$^P$-binding proteins other than Bub1–Bub3. In this study, we demonstrate that Bub1–Bub3 binds directly to MELT$^P$ sequences and recapitulate the checkpoint defect by mutating residues within the MELT$^P$-binding site of Bub3 that impair its own kinetochore recruitment and that of Bub1.

Overall, these observations indicate that Bub1 localization to kinetochores is required for checkpoint function. We suspect that the recruitment of Bub1–Bub3 to Spc105/Knl1 unleashes Bub1's role in the checkpoint. Because the kinase domain of Bub1 is not required for checkpoint signaling (*Sharp-Baker and Chen, 2001*; *Fernius and Hardwick, 2007*; *Perera et al., 2007*; *Klebig et al., 2009*; *Ricke et al., 2012*), Bub1's role in the checkpoint is probably entirely exerted through macromolecular interactions. For instance, Bub1 is required for kinetochore recruitment of Mad1-Mad2 and of Mad3/BubR1 (*Millband and Hardwick, 2002*; *Liu et al., 2006*; *Rischitor et al., 2006*; *Perera et al., 2007*; *Logarinho et al., 2008*; *Klebig et al., 2009*) (*Figure 6F*).

Mad3/BubR1 is, like Bub1, constitutively bound to Bub3. Thus, a requirement on Bub1 for kinetochore recruitment of Mad3/BubR1 is unexpected and suggests that when bound to Mad3/BubR1, Bub3 might be unable to perform an equivalent function in kinetochore recruitment to the one it performs when bound to Bub1. Indeed, our studies indicate that when bound to Bub3, Bub1 plays an important positive role in the interaction with MELT$^P$ (*Figure 4D–F*). Although the presence of a positively charged residue at position 314 is not a conserved feature of Bub1 (not shown), it is plausible that differences in the sequence of the β1-β2-loop in Bub1 and Mad3/BubR1 (*Figure 2D*) might account for the different behavior of Bub3 in its respective complexes.

Multisite phosphorylation is commonly used in signaling networks for its potential to generate nonlinear responses to stimuli (*Kapuy et al., 2009*; *Salazar and Höfer, 2009*). Multisite phosphorylation of Sic1 and Cyclin E by Cyclin-dependent kinases, for instance, mediates their interaction with the Cdc4 and Fbw7 ubiquitin ligases and their subsequent ubiquitination and destruction (*Nash et al., 2001*; *Orlicky et al., 2003*; *Hao et al., 2007*; *Kõivomägi et al., 2011*). Similar to Bub3, also Cdc4 and Fbw7 are WD40 β-propeller proteins, but the position and organization of the phosphopeptide binding sites in Cdc4 and Fbw7 and in Bub3 are distinct. In Cdc4 and Fbw7 (as well as in β-TrCP, another WD40 β-propeller Ub-ligase that interacts with phosphorylated motifs in target proteins) the phosphopeptide-binding site is located on the top surface of the propeller domain (*Figure 3—figure supplements 1 and 2*) (*Orlicky et al., 2003*; *Wu et al., 2003*; *Hao et al., 2007*). The binding affinity of Bub1–Bub3 for individual MELT$^P$ repeats of Spc105/Knl1 is high, matching the highest binding affinities measured for SH2-phosphopeptide interactions (*Seet et al., 2006*). This suggests that each MELT$^P$ sequence has the potential to act as a docking site for an individual Bub1–Bub3 complex and that multiple Bub1–Bub3 complexes may bind to Spc105/Knl1 concomitantly if multiple phosphorylated MELT repeats are present (*Figure 6F*). Understanding how variations in the levels of phosphorylation of MELT motifs reflect on the strength of the checkpoint response is a crucial question for future studies.

The high affinity of the Bub1–Bub3 complex for MELT$^P$ sequences might explain why Bub1 is quite stably bound to kinetochores during checkpoint activation, as shown by fluorescence recovery after photobleaching (FRAP) experiments (*Howell et al., 2004*; *Shah et al., 2004*). It is possible that additional interactions of the Bub1–Bub3 complex with Spc105/Knl1 or other checkpoint components at the kinetochore further increase the binding affinity of the Bub1–Bub3 complex for kinetochores. The TPR domain of Bub1, whose function is essential for the spindle checkpoint (*Kadura et al., 2004*; *Vanoosthuyse et al., 2004*; *Kiyomitsu et al., 2007*; *Klebig et al., 2009*), might be playing a crucial role in such additional interactions of Bub1 (*Brady and Hardwick, 2000*; *Kiyomitsu et al., 2007*; *Klebig et al., 2009*; *Bolanos-Garcia et al., 2011*; *Kiyomitsu et al., 2011*; *Kim et al., 2012*; *Krenn et al., 2012*). In this study, we have clarified how Bub1 becomes recruited to kinetochores and shown that such recruitment is crucial for the spindle checkpoint. Future studies will have to shed light on the molecular mechanisms subtending the function of Bub1 at kinetochores and shift the focus to asking how the recruitment of Bub1 leads to the recruitment of Mad1 and Mad2 and the generation of a diffusible signal that emanates from the kinetochore.

## Materials and methods

### Protein expression and purification

cDNA sequences coding for *S. cerevisiae* Bub1$^{289–359}$ and for full length Bub3 were subcloned in the first and second cassettes of pGEX-6P-2rbs vector (*Ciferri et al., 2005*). In this construct, the cDNA

encoding Bub1[289–359] was sub-cloned in frame with the gene encoding GST and a cleavage site for PreScission protease, and was translated from the first ribosome-binding site (rbs). Untagged Bub3 was subcloned downstream of the second rbs in the same vector. Expression was carried out in BL21 DE3 plysS cells in LB medium by auto-induction with 0.3% lactose at 18°C for approximately 16 hr. Cells were harvested by centrifugation and resuspended in Buffer A (20 mM Tris pH 7.5, 300 mM NaCl, 10% Glycerol, 1 mM DTE, 1 mM EDTA pH 8.0, 1 mM PMSF) typically at a dilution of 3 ml lysis buffer per ml of bacterial pellet. Cells were lysed by sonication, and the lysates were cleared by centrifugation at 75000 × g for 45 min. The resulting clear supernatant was incubated with gentle rotation with 1/50 (vol/vol) of GSH-sepharose slurry (GE Healthcare) for 1–2 hr at 4°C. Beads were washed with 150 vol of Buffer A. To elute the Bub1–Bub3 complex from beads, GST-PreScission protease (0.02 mg/mg of protein target) was added for 16 hr at 4°C. The eluate was concentrated on Amicon Ultra Centrifugal Filters MwCO 3000 (Merck Millipore, Billerica, MA) and further purified by size exclusion chromatography (SEC) on a Superdex 75 16/60 column using GF buffer (20 mM Tris pH 7.5, 150 mM NaCl, 1 mM DTE). The complex eluted in a single peak with apparently stoichiometric amounts of Bub1[289–359] and Bub3 and was subsequently concentrated to ~10 mg/ml for crystallization or to 2–4 mg/ml for ITC experiments. The MELT2[P] (sequence DPTSMEM{PTHR}EVFPRSIRQKN), MELT2 (DPTSMEMTEVFPRSIRQKN), MELT4[P] (DTVEGEPIDL{PTHR}EYESKPYVPN), and MELT4 (DTVEGEPIDLTEYESKPYVPN) peptides (95% purity) were custom made by GeneScript (Piscataway, NJ). Histone H1-derived phosphorylated peptide (GGGPA{pTHR}PKKAKKL, 95% purity) was purchased from AnaSpec (Catalog number 61741).

## ITC measurements

Binding isotherm was measured at 25°C by isothermal titration calorimetry on a MicroCal ITC200 device (GE Healthcare, Piscataway, NJ). All samples were extensively dialysed into fresh GF buffer (20 mM Tris pH7.5, 150 mM NaCl, 1 mM DTE). In each titration, the Bub1[289–359]–Bub3 in the cell (at a 30 µM concentration) was titrated with thirty-five 2-µl injections (at 90 s intervals) of the indicated synthetic MELT[P] peptides (at a concentration of 400 µM). The injections were continued beyond saturation levels to allow for determination of heats of ligand dilution. Data were fitted by least-square procedures to a single-site binding model using ORIGIN 5.0 software package (MicroCal, Northampton, MA).

## Strains, media and reagents

All yeast strains (*Table 2*) were derivatives of, or were backcrossed at least three times to, W303 (ade2-1, trp1-1, leu2-3, 112, his3-11, 15, ura3, ssd1). Cells were grown in YEP medium (1% yeast extract, 2% bactopeptone, 50 mg/l adenine) supplemented with 2% glucose (YEPD). α-factor and nocodazole were used at 3 µg/ml and 15 µg/ml, respectively. 150 minutes after α-factor release, nocodazole was re-added to the cultures at 7.5 µg/ml. Synchronization experiments were carried out at 30°C.

**Table 2.** Strains used in this study (all in W303 background)

| Name | Relevant genotype |
| --- | --- |
| yAC1 | *MATa* |
| yAC411 | *MATa, bub3::LEU2* |
| yAC1990 | *MATa, bub3::LEU2 trp1:: BUB3[R127A/R239A]-3Cherry::TRP1* |
| yAC2036 | *MATa, bub3::LEU2 trp1:: BUB3[R239A]-3Cherry::TRP1* |
| yAC2048 | *MATa, bub3::LEU2 trp1:: BUB3[R127A/R239A]-3Cherry::TRP1* |
| yAC2072 | *MATa, bub3::LEU2 trp1:: BUB3[R127A/R239A]-3Cherry::TRP1, MTW1-GFP::TRP1* |
| yAC2110 | *MATa, bub3::LEU2 trp1::BUB3-3Cherry::TRP1, MTW1-GFP::TRP1* |
| yAC2090 | *MATa, bub3::LEU2 trp1::BUB3-3Cherry::TRP1, BUB1-GFP::TRP1* |
| yAC2091 | *MATa, bub3::LEU2 trp1:: BUB3[R127A/R239A]-3Cherry::TRP1, BUB1-GFP::TRP1* |

## Plasmid constructions and genetic manipulations

To obtain strains expressing Bub3-mCherry or Bub3[R127A–R239A]-mCherry, plasmids AC122 and AC125 were created by subcloning in Yiplac204 cDNA fragments encoding wild type Bub3 or the double mutant (with 250 bp upstream of ATG and 200 bp downstream of the stop codon). The Bub3 sequences were fused to three copies of mCherry that had been amplified from pCM79-pFA6a::3mcherry::hphNT1 (*Maeder et al., 2007*). The plasmids were integrated at the TRP1 locus by digestion with Bsu36I, and the copy number of the integrated plasmids was verified by Southern blotting.

## Flow cytometry and other techniques

Flow cytometric DNA quantitation was performed on a Becton-Dickinson (Franklin Lakes, NJ) FACScalibur device and analysed with CellQuest software. Kinetics of re-budding was scored on ethanol-fixed cells.

## Live cell imaging

Time lapse videos were performed at 30°C using CELLASIC microfluidic chambers and recorded using a Delta Vision Elite imaging system (Applied Precision, Issaquah, WA) based on an IX71 inverted microscope (Olympus, Shinjuku, Tokyo, Japan) with a CoolSNAP HQ2 camera (Photometrics, Tucson, AZ) and a UPlanApo 60 × (1.4 NA) oil immersion objective (Olympus).

## Crystallization and structure determination

Prior to crystallization, the Bub1[289–359]–Bub3 complex was mixed at a 1:2 ratio with a synthetic Spc105/Knl1 phosphopeptide (sequence DPTSMEM{T$^P$}EVFPRSIRQKN, with N-terminal amide and C-terminal acetyl groups) and subjected to crystallization by the sitting drop method with a Mosquito nanodrop dispenser. Initial crystals were obtained with the G9 condition of Qiagen (Venlo, The Netherlands) PACT Suite screen (0.2 M K/Na tartrate, 0.1 M Bis Tris propane pH 7.5, 20% PEG 3350) and did not require further optimization. The crystals grew to a typical size of ~50 µm in each direction. X-ray diffraction data were collected at the PXII–X10SA beamline at the Swiss Light Source (SLS) (Villigen, Switzerland) and processed using XDS (*Kabsch, 2010*). Due to anisotropic diffraction, the data were subject to anisotropy correction using the UCLA diffraction anisotropy server (*Strong et al., 2006*). Model refinement against the corrected data resulted in final maps of significantly better quality compared to maps obtained with uncorrected data. Initial phases were obtained by the molecular replacement method, which was carried out using the program PHASER (*McCoy et al., 2007*) and the structure of *S. cerevisiae* Bub3–Bub1 (PDB ID 2I3S) (*Larsen et al., 2007*) as a search model. Two copies of the Bub3–Bub1 dimer were placed in the asymmetric unit. Model building and refinement were carried out using Coot (*Emsley et al., 2010*) and phenix.refine from the PHENIX suite (*Adams et al., 2010*), respectively. The final model contains two Bub3 monomers (including residues 1–223 and 233–340 in chain A and residues 1–223 and 234–340 for chain D), two Bub1 fragments (including residues 302–311 and 314–347 for chain B and residues 309–347 for chain E), and two Spc105/Knl1 MELT$^P$ peptides (including residues 165–176 in chain C and residues 166–177 in chain F). Simulated annealing composite omit maps were produced using phenix.autobuild (*Terwilliger et al., 2008*). Figures were generated using either CCP4MG (*McNicholas et al., 2011*) or Pymol (Schrödinger LLC, Portland, OR). The final model and the structure factor amplitudes have been submitted to the Protein Data Bank under the accession numbers 4bl0 and r4bl0sf, respectively.

## Localization index

Segmentation and fluorescence analysis for single cell images was performed with software written in MATLAB. For the segmentation and tracking of yeast cells, we used the program 'phyloCell', written by Gilles Charvin (unpublished results). To quantify the localization of proteins, we focused on the brightest pixels within each segmented area (i.e., within each cell). We observed that the area where the brightest pixels are typically localized amounts to roughly 1% of the area of the whole cell. Therefore, to compute an index for localization, we calculated the average of the brightest 1% of pixels. The value of this average depends on properties of the overall intensity distribution. When confronting two Gaussian distributions of pixel intensities, it is expected that the value of the average of the brightest intensity will be higher for the distribution whose mean intensity is higher or alternatively, for distributions with similar mean intensity, for the distribution whose standard deviation is higher. To correct for these factors, we used the following measure to quantify localization:

Localization index = ((average 1% brightest pixels) − mean)/std

This definition allows us to distinguish quantitatively between localization and delocalization. In case of perfect delocalization, the overall intensity will adopt a gaussian distribution, and in this case our measure of localization adopts a value of ~3.7 (*Figure 5—figure supplement 2A* for an example of overall intensity distribution when the protein is delocalized and for the corresponding localization index). When the protein is kinetochore-localized (as shown by co-localization with Mtw1), the distribution is skewed-Gaussian, with a more extended right tail (*Figure 5—figure supplement 2B* for an example of overall intensity distribution when the protein is kinetochore localized and for the

corresponding localization index). In this case, our measure for localization is larger than ~3.7. When the localization index assumes values that are significantly higher than ~3.7, we observed the fluorescent protein to be localized to kinetochores, as confirmed by co-localization with Mtw1 (*Figure 5*).

The four plots in *Figure 5—figure supplement 2C–E* report the distribution of maxima (C), minima (D), and standard deviation (E) of the localization index of Bub3$^{wt}$-mCherry or Bub3$^{R217A-R239A}$-mCherry over an entire cell cycle (i.e., the time between two budding events). For Bub3$^{wt}$-mCherry, we measured 55 cycles in 37 cells. For Bub3$^{R217A–R239A}$-mCherry, we measured 45 cycles in 38 cells. The distribution of minima of the localization index is similar for the two Bub3 species, corresponding to delocalization. Wild type and mutant, however, differ clearly for the distribution of maxima, where the wild type only shows localization (i.e., localization index >3.7). The higher amplitude of the localization index in wild types gives rise to a larger standard deviation of the localization index when compared to the wild type (C).

## Acknowledgements

We are grateful to Jonathan Millar, Andrew Murray, Veronica Krenn, Katharina Overlack and Clauda Breit for critical reading of the manuscript and to members of the Musacchio laboratory for helpful discussions. We thank Ingrid Vetter, Marco Bürger and the staff of the PSI for precious help in X-ray diffraction data collection, and Peter De Wulf and Simonetta Piatti for sharing reagents.

## Additional information

### Funding

| Funder | Grant reference number | Author |
| --- | --- | --- |
| European Research Council | 233316 (KINCON) | Andrea Musacchio |
| Seventh Framework Programme of the European Commission | Integrating Project MitoSys | Andrea Musacchio |
| Italian Association for Cancer Research (AIRC) | | Andrea Ciliberto |
| Umberto Veronesi Foundation | | Elena Chiroli |

The funders had no role in study design, data collection and interpretation, or the decision to submit the work for publication.

### Author contributions

IP, Conception and design, Acquisition of data, Analysis and interpretation of data, Drafting or revising the article; JRW, EC, Acquisition of data, Analysis and interpretation of data, Drafting or revising the article; FG, Analysis and interpretation of data, Drafting or revising the article; IH, SvG, Acquisition of data; AC, AM, Conception and design, Analysis and interpretation of data, Drafting or revising the article

## Additional files

### Major dataset

The following dataset was generated:

| Author(s) | Year | Dataset title | Dataset ID and/or URL | Database, license, and accessibility information |
| --- | --- | --- | --- | --- |
| Weir J, Primorac I, Musacchio A | 2013 | Crystal structure of yeast Bub3-Bub1 bound to phosopho-Spc105 | 4bl0; http://www.rcsb.org/pdb/search/structidSearch.do?structureId=4bl0 | Publicly available at the RCSB Protein Data Bank (http://www.rcsb.org/). |

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
