## [Decision Letter]

Thank you for submitting your work entitled “Bub3 reads phosphorylated MELT repeats to promote spindle assembly checkpoint signaling” for consideration at *eLife*. Your article has been favorably evaluated by a Senior editor and three reviewers, one of whom is a member of the Board of Reviewing Editors.

One referee would like you to perform the following experiments to support your conclusions:

1) Arg239 that contacts pThr is a Lys in most Bub1 orthologs. Did the authors test whether a Lys substitution can still bind MELT peptides?

2) The authors suggest that Arg314 mediates the Bub1-dependent increased affinity of Bub3 for MELT peptides. What happens to the Bub1-mediated increase in binding when Arg314 is mutated?

---

## [Author Response]

*1) Arg239 that contacts pThr is a Lys in most Bub1 orthologs. Did the authors test whether a Lys substitution can still bind MELT peptides*?

[Editors' note: at the authors' request, the answer to this question has not been published to avoid disclosing confidential information.]

*2) The authors suggest that Arg314 mediates the Bub1-dependent increased affinity of Bub3 for MELT peptides. What happens to the Bub1-mediated increase in binding when Arg314 is mutated*?

This was an interesting suggestion. We have carried out an additional calorimetry experiment with the Bub1^R314A^ mutant. In agreement with the hypothesis that this residue contributes to recognition of the phosphorylated MELT peptide, we find that the R314A mutant binds the MELT2^P^ peptide with ∼3- to 4-fold reduced affinity. We have included the new ITC data as Figure 4.